# Payoff information hampers the evolution of cooperation

Steffen Huck[1,2], Johannes Leutgeb[1] & Ryan Oprea[3]

Human cooperation has been explained through rationality as well as heuristics-based models. Both model classes share the feature that knowledge of payoff functions is weakly beneficial for the emergence of cooperation. Here, we present experimental evidence to the contrary. We let human subjects interact in a competitive environment and find that, in the long run, access to information about own payoffs leads to less cooperative behaviour. In the short run subjects use naive learning heuristics that get replaced by better adapted heuristics in the long run. With more payoff information subjects are less likely to switch to pro-cooperative heuristics. The results call for the development of two-tier models for the evolution of cooperation.

[1] WZB, Reichpietschufer 50, 10785 Berlin, Germany. [2] Department of Economics, University College London, Gower Street, London WC1E 6BT, UK. [3] Department of Economics, University of California, Santa Barbara, 3014 North Hall, Santa Barbara, California 93106-9210, USA. Correspondence and requests for materials should be addressed to S.H. (email: steffen.huck@wzb.eu).

Grasping the forces and conditions that foster cooperative behaviour is of central importance for the understanding of how societies thrive or decline. Cooperation has been explained through models with purely selfish agents who exhibit high levels of sophistication and reasoning in repeated-game environments[1,2]; static models invoking social preferences, altruistic punishments and reciprocity[3–11]; evolutionary models that show under which conditions cooperative behaviour, reciprocity and pro-social preferences survive[12–16]; and models showing how learning rules and heuristics such as win–stay, lose–shift or win–continue and lose–reverse can generate cooperative behaviour[17–20]. All these models either depend on the assumption that agents have full knowledge of their own 'payoff function' (that is, that they know how all possible outcomes in the game they play map into success), or, where such knowledge is not required, imply that having access to payoff information cannot hurt.

Here we present evidence to the contrary: we show that in the long run, learning environment payoff information hampers the evolution of cooperation. Our evidence suggests that, for understanding behaviour in the long run, two-tier models combining elements of evolutionary models and models of learning rules and heuristics are appropriate. In the short run, we find that subjects' behaviour is best described by naive learning heuristics but in the long run these heuristics are replaced by better adapted heuristics. We show that subjects are much less likely to learn to adopt heuristics that foster cooperation if they have access to payoff information.

## Results

**The experiment.** We study learning behaviour in a game with 600 periods that each last 8 s. (We also have data on games with 1,200 4-s and 2,400 2-s periods that generate similar results. See Supplementary Figs 1 and 2 as well as Supplementary Discussion.) The game is a two-player contest game with a continuous action space that exhibits a strong tension between competition and cooperation. The payoff function is

$\pi_i(x_i, x_{-i}) = 10 + \left(\frac{120}{\sum_j x_j} - 10\right)x_i$ where $x_i$ denotes the action of

agent $i$; maximum competition implies $x_i = 6$ with profits of 0.5 euro cents per period, maximum cooperation $x_i = 0.1$ with profits of 3.45 cents per period and Nash in between at $x_i = 3$ and profits of 2 cents per period. Points are converted into Euro by dividing them by 2,000. The one-period Nash equilibrium outcome is in the middle of the action space with an effort of 3 and profits of 2 cents. There are two treatments, Info and NoInfo, varying the feedback information that subjects receive between rounds. In both treatments, subjects observe the action profile of the last period and the resulting payoffs for themselves and their partner. In treatment Info they additionally observe a curve showing the possible payoffs they would have earned in the previous period for alternative choices. Note that the NoInfo treatment is based on previous work by Friedman et al.[21] that use the same software package albeit slightly different parameters with 4 s per period and rematching of subjects every 400 periods. Our findings successfully replicate their findings.

Running this many periods of interaction in a single laboratory session is made feasible using a graphical software interface (see Fig. 1) where choices are implemented through a slider. Actions are determined by the slider's position at the end of the timed period and the slider remains in the same position when the next period starts. Subjects are shown a progress bar filling up every 8 s and a counter of remaining periods. In the first period sliders are set at a random position that determines play in that period. After a period has ended subjects receive feedback on both

players' actions and payoffs in the previous period: two dots in the interface indicate (through their horizontal position) actions and (through their vertical position) payoffs. Furthermore, in treatment Info subjects are shown a curve of potential payoffs that they could have earned given their partner's choice. Notice that subjects are, of course, free to leave the slider in the same position for multiple rounds, in which case actions from previous periods are simply repeated. Indeed, many subjects often choose to wait a little between adjustments, which renders the game much less volatile and stressful than the sheer number of periods might suggest.

**Aggregate results.** Figure 2 shows median actions over the course of the experiment. The experimental results can be divided into three phases: first, a phase in which subjects employ naive heuristics shaped by salient feedback information; second, a learning phase in which subjects revise their initial naive heuristics; and, third, a long-run phase where behaviour settles down. In what follows we separate the phases at period 25 and period 300. All results are robust to changing the separating periods by ± 20%.

Information provided has a strong influence on which naive heuristics subjects initially choose. In the naive NoInfo heuristics phase, subjects choose actions above the Nash prediction—the often-observed consequences of imitate-the-best heuristics that subjects frequently employ in low-information settings in previous experiments[21–24]. In contrast, during the same initial naive phase, subjects in the Info treatment rapidly converge on the static Nash equilibrium outcome. This behaviour is consistent with a myopic best-reply heuristic, also documented in previous studies in which payoff information was available[21–26]. The difference between the two treatments is significant (two-sided Mann–Whitney U-test, $P < 0.05$).

After 25 periods, subjects in NoInfo dramatically decrease their actions, indicating that they successfully abandon the imitation heuristic. Subjects in Info exhibit a much weaker downward

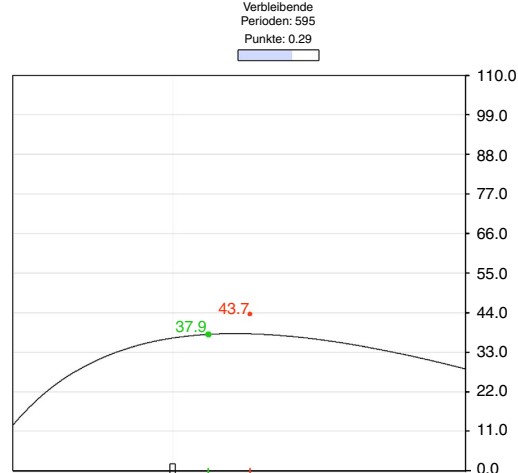

**Figure 1 | Interface in the Info treatment.** The subject chooses her action on the horizontal axis via a slider. The green and red dots represent a subject's, respectively his or her partner's, earnings in the preceding period. The black line shows the potential payoffs in the preceding period. On the top of the interface, the subjects see the number of remaining periods, the amount of points they earned and how much time they have left to decide via a progress bar. The interface in NoInfo is exactly the same except for the black curve of potential payoffs. Note that, in the experiment, the x axis is intentionally left unlabelled to avoid giving the subjects any cues on how to behave. The y axis shows the payoffs in points.

pattern, suggesting that they have a harder time abandoning the myopic best-reply heuristic. From period 300 on, behaviour stabilizes in the long-run phase. In NoInfo subjects choose actions that are significantly more cooperative than the Nash outcome (two-sided $t$-test, $P<0.001$), whereas in Info their average action is indistinguishable from the Nash outcome (two-sided $t$-test, $P>0.1$). Crucially, subjects cooperate much more successfully in treatment NoInfo than in Info (two-sided Mann–Whitney $U$-test, $P<0.001$), generating also significantly higher payoffs in the long-run phase. In the final phase median earnings are 6.01 Euro in Info, whereas in NoInfo they are over 50% higher at 9.56 Euro. Note that this pattern is replicated in the treatments with 1,200 and 2,400 periods, although subjects in the Info treatments show slightly more collusive behaviour. See Supplementary Figs 1 and 2 as well as the Supplementary Discussion.

**Individual behaviour.** What individual behavioural patterns give rise to the differences in aggregate behaviour documented above? To find out, we run simple regressions to investigate whether the changes that we observe are consistent with a set of heuristics relevant in our setting. Specifically, we examine 'Imitate-the-Best', which tells a subject to copy the opponent's action in the previous period if the opponent chose a higher action and made higher profits; 'Match', which tells a subject to copy the other player's action regardless of payoff; 'Win-Continue, Lose-Reverse' (WCLR) where a subject repeats the previous round's adjustment if it improved his or her payoff and changes directions if profits declined; and, finally, 'Myopic Best Reply', which tells subjects to move towards the best reply against their opponent's current action. While imitate-the-best pushes behaviour towards extremely competitive outcomes (a higher action always generates a higher relative payoff)[27] and myopic best replies push behaviour towards Nash[28], both, the Match heuristic and WCLR, are heuristics that foster cooperation. As with tit-for-tat, Match rewards an opponent's move towards cooperative play and punishes deviations to higher actions. Moreover, Match aligns subjects' actions thereby speeding up WCLR as an effective force for pushing actions towards cooperation[13,19,20]. Obviously, Myopic Best Reply is easily available for subjects in the Info treatment, whereas it requires information acquisition and much more sophistication in the NoInfo treatment.

Some of these heuristics give point predictions whereas others yield predictions about the direction of change. To make comparable comparisons, we focus only on changes in decisions relative to the subject's previous period action. The outcome variable CHANGE gives the direction of change. Independent variables COPY UP and COPY DOWN capture the Imitate-the-Best and Match heuristics. Finally, WCLR is captured by the variable of the same name and Myopic Best Reply is captured by variable BR. The treatment effects are captured by treatment dummy INFO. We also interact treatment dummy INFO with all of the heuristics variables described above to pick up the differences in subjects' use of the modelled heuristics. The exact variable definitions are reported in the Methods section.

Table 1 reports the results of the model estimated separately on the naive, and then the learning and long-run phases combined. The first column reports estimates for the naive phase, that is, the first 25 periods. The coefficient COPY UP is significant and large, whereas COPY DOWN is insignificant. This indicates strong imitate-the-best behaviour in NoInfo. In Info, on the other hand, there is no evidence of imitate-the-best behaviour. The interaction term COPY UP:INFO is negative and weakly significant and cancels out the baseline effect entirely (Wald test $P>0.1$). In NoInfo the BR coefficient picks up some regression to the mean, while the large and highly significant coefficient for the interaction term BR:INFO indicates just how appealing subjects find the best-reply heuristic once they have the relevant information. Finally, there is some weakly significant evidence for WCLR already in the first phase. Some subjects may already switch from imitation to WCLR at the end of the initial phase.

Column 2 reports estimates for the learning and the long-run phases (periods 25 to 600). In NoInfo, COPY UP remains significant while COPY DOWN now also turns significant and negative. There is therefore a switch from Imitate-the-best to the Matching heuristic. The estimations also show that WCLR becomes a key heuristic in NoInfo. This behaviour gives rise to the decrease in actions in the learning phase and stabilizes cooperative play in the long-run. The interaction term WCLR:INFO is negative if insignificant, and the linear combination of baseline and interaction term is insignificant

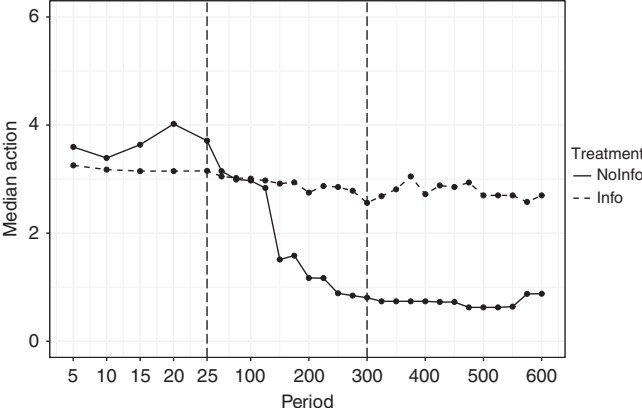

**Figure 2 | Median actions.** Up to period 25, each dot represents the median action in bins of 5 periods. From period 25 (dashed vertical line), on each dot represents the median action in bins of 25 periods. In the naive phase (periods 1–25), the average action (s.d. in parentheses) is 3.65 (1.58) in NoInfo and 3.37 (1.19) in Info. In the learning phase (periods 26–300), this changes to 2.22 (1.90) in NoInfo and 2.93 (1.65) in Info. Finally, in the long-run phase (periods 301–600), behaviour settles down at 1.23 (1.45) in NoInfo and 2.62 (1.80) in Info.

**Table 1 | Linear probability model estimates.**

| | Dependent variable | |
|---|---|---|
| | **CHANGE** | |
| | **(1) Periods 1–25** | **(2) Periods 26–600** |
| COPY UP | 0.422***(0.144) | 0.274*** (0.077) |
| COPY DOWN | 0.015 (0.172) | − 0.221** (0.082) |
| WCLR | 0.062* (0.031) | 0.108** (0.049) |
| BR | 0.114*** (0.037) | 0.087* (0.047) |
| INFO | 0.355* (0.186) | − 0.196* (0.109) |
| COPY UP:INFO | − 0.518*** (0.168) | 0.124 (0.106) |
| COPY DOWN:INFO | − 0.362* (0.196) | 0.285** (0.109) |
| WCLR:INFO | − 0.006 (0.053) | − 0.065 (0.057) |
| BR:INFO | 0.226*** (0.066) | 0.111* (0.062) |
| Constant | 0.264 (0.161) | 0.347*** (0.082) |
| Observations | 1,428 | 26,216 |
| $R^2$ | 0.233 | 0.250 |
| F-statistic | 47.91*** | 81.48*** |

We report a linear probability model estimated by ordinary least squares (OLS) as the interpretation of the coefficients, especially of the interaction terms, is most straightforward. The results of a Logit estimation are qualitatively similar. To check robustness we also run regressions where we drop observations on the limit of the action space as subjects can then only increase respectively decrease their action. The results remain qualitatively robust. Variable definitions are reported in the Methods section. Note that interaction terms are denoted by a colon. In this regression we pool data from learning and long-run phases as results do not change qualitatively between them. Supplementary Table 1 reports separate regressions for learning and long-run phases.
*$P<0.1$, **$P<0.05$, ***$P<0.01$; s.e. clustered on groups in parentheses.

(Wald test $P > 0.1$). Thus, there is no evidence of subjects following WCLR in the Info treatment. Instead, the interaction effect BR:INFO is weakly significant and when combined with the baseline is highly significant (Wald test $P < 0.01$), indicating that subjects continue strongly to follow the Myopic Best Reply heuristic. These results thus show that subjects find it much harder to abandon their initially chosen heuristic in Info.

## Discussion

Payoff information can hurt because it can affect the initial heuristics that subjects become attached to as well as subjects' long-run affinity to that heuristic. This is in stark contrast to rational models of cooperation. The results strongly suggest that subjects lean heavily on heuristics instead of employing more sophisticated repeated-game logic to achieve cooperation in early play as much as in the long run. Most importantly, dropping superficially plausible heuristics like myopic best reply (which maximizes short-run payoffs) turns out to be much harder than dropping a more obviously ill-adapted heuristic like imitate-the-best (that generates much lower payoffs by inducing extreme competition). As a result, payoff information is an advantage only in the short run, hampering the learning necessary to establish cooperation in the long run. Modelling such long-run evolution of cooperation requires a two-tier model where subjects follow heuristics for a while but adapt or replace them over time. Fully accounting for such long-run cooperative evolutionary processes requires models that describe not only the heuristics employed by agents in an interaction but also how and when agents learn to improve these heuristics over time.

Finally, some thoughts on external validity are in order. In our laboratory setting feedback on others' actions and profits and feedback on possible payoffs that could have been achieved are extremely salient and there are no information channels that would connect individuals to others outside their immediate group of competitors. We would, of course, expect that learning reacts to changes in the environment such as readily available multi-period memory, information from other groups of players or endogenous choice of partners and evolutionary selection. Indeed, incorporating such richer information and social structures provides many interesting possibilities for future research. However, we believe that when it comes to the external validity of our main result we can argue that our Info treatment provides an ideal setting for payoff information to aid subjects' rationality. There is very little distraction and experimentation to learn more about the payoff landscape remains, thanks to the large number of repetitions, extremely cheap. If human cooperation were largely driven by rational calculation, our Info treatment should clearly have speeded up cooperation and should have increased subjects' earnings throughout the experiment. We find evidence to the contrary despite the ideal conditions provided in the laboratory.

## Methods

**Experimental setup.** The experiment was run employing the ConG software package[29]. In each of the two treatments we observe 18 independent anonymous pairs of subjects who play the game for 600 periods. Subjects were recruited via ORSEE[30] at the WZB-TU laboratory in Berlin. Sessions lasted 80 min and subjects' average payouts were 12.63 Euro in Info and 15.66 Euro in NoInfo, plus a 5 Euro show-up fee. Upon arrival subjects were randomly allocated to seats and received written instructions explaining (1) they would play against one other participant; (2) how they would choose from their action spaces; and (3) what feedback they would receive through the graphical interface. Subjects are not made aware of the payoff function. They only know that the payoff function depends exclusively on both players' choices and that it is symmetric and does not change over time. The experimental instructions are reported in the Supplementary Methods.

**Variable definitions.** We define the variables in the regression in Table 1 as follows. The outcome variable CHANGE is coded as 1 for an increase and 0 for a decrease in action. Imitate-the-best is captured by the COPY UP dummy that is coded as 1 if the opponent chose a higher action in the previous period. The match heuristic is covered by both COPY UP and COPY DOWN (that is coded as 1 if the opponent had a lower action). WCLR is captured by the dummy of the same name that is coded as 1 if the heuristic prescribes an increase and 0 otherwise. For constant profits, we code the variable as 1. The results are robust to coding it as 0 or dropping these observations. Finally, myopic best-replies are covered by the BR dummy that is coded as 1 when the myopic best reply was above a subject's previous period action and 0 otherwise. There are no observations for which the myopic best-reply would be exactly zero. Finally, we have a treatment dummy INFO that is 1 for the Info treatment and 0 for NoInfo.

**Code availability.** The computer code used in the analysis is available from the corresponding author on request.

**Data availability.** The data that support the findings of this study are available from the corresponding author on request.

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

## Acknowledgements

Support by German Science Foundation through CRC TRR 190 is gratefully acknowledged.

## Author contributions

All authors contributed to all aspects of this work.

## Additional information

**Competing interests:** The authors declare no competing financial interests.

