## [Peer Review File · Nature Communications]

Reviewer #1 (Remarks to the Author):

The authors conduct a two-player repeated game experiment to examine the consequences that two different payoff information scenarios for the participants have on their individual behaviour. In the first scenario (called NoInfo), participants are told the choices (in the interval $[0.1, 6]$) and payoffs of the two players in the previous period, while in Info they are also shown graphically what their payoff would have been in the previous round for all their choices given the actual choice of their partner.

The results show that participants tend to follow a myopic best reply heuristic (BR) leading to the unique Nash equilibrium (NE) of the one-shot game when the necessary information is available (i.e. in Info). However, in NoInfo, participants tend to follow an Imitate-the-Best heuristic for the first 25 periods or so and, over the remainder of the 600 periods, switch to a combination of adopting their partner's previous choice (Match) and adjusting their choice in the same (respectively, opposite) direction if their payoff increased (respectively, decreased) in the previous period (WCLR).

The game's payoff function is such that Imitate-the-Best leads to choice 6 with lowest payoff for both players, the unique NE is choice 3 with intermediate payoff, and Match combined with WCLR leads to choice 0.1 with highest payoff. Thus, the extra payoff information in Info produces a less cooperative outcome in the long-run, hence the title for the paper. For a different payoff function, one can well imagine the above behavioural heuristics would produce the opposite title.

The properties of the one-shot payoff function, now briefly mentioned on page 7, need to be discussed earlier in the paper. This will allow readers to understand such statements on page 4 as Imitate-the-Best copies "a higher action and made higher profits" and "pushes behaviour towards extremely competitive outcomes". Moreover, more discussion of reference 24 (Friedman et al., 2015) and its connection to the current paper must be provided in the main text since its two-player experiment seems to be based on the Info treatment with twice as many (i.e. 1200) periods. The data the authors have on 1200 and 2400 period games should be given in an Appendix. More statistics should be provided for Figure 2 (e.g. means and variances) as well as extra columns in Table 1 for Periods 26-300 and Periods 300-600. Again, these can be relegated to an Appendix if necessary. Also, in Table 1, it is unclear to me whether the first four rows refer to data from only the treatment NoInfo. If the data is for the combined NoInfo and Info treatments, I think extra rows are needed for only data from NoInfo.

Overall, the paper shows clearly that details of payoff information available to the players has important consequences in predicting their behaviour, a valuable lesson given the recent proliferation of experiments based on more complicated games. I would like to see a revision that addresses my concerns above, including the relationship with reference 24.

Minor Comments. P4: Could mention "Myopic Best Reply" is only available in Info.

P4, line 11-: How does "Match align subjects' actions" rather than produce a two-cycle?

P6, line 1-: Perhaps "unchanging over time" is better than "stable over time"?

Reviewer #2 (Remarks to the Author):

Huck et al. has subjects play a many round repeated game with a continuous action space where subjects learn their payoffs from each round as well as the payoffs the other player receives, and depending on the treatment, also learn the payoffs they could have received for every action they could have taken. Subject of course do not know the game form, i.e. the complete mapping between the action profile and the payoffs. The authors find the learning models that best fit the subjects data, and also analyze the long run behavior of subjects in each treatment, and

convincingly demonstrate that subjects in the treatment with less payoff information perform better, presumably because they are less likely to get stuck in the local optima that consists of the Nash equilibrium of a single stage in the game but more likely to achieve the global optima represented by a more cooperative equilibria of the repeated game, presumably because when the subjects have more information they are more likely to employ as their learning process the heuristic of best response dynamic.

I find this result to be compelling, and the design to be rather clean, and fits in well with the learning and games literature, and also seems to fit a generally valid intuition that having immediate payoffs can make it more likely to get stuck at a local optima and prevent people from realizing long term gains, which can be crucial in repeated games settings. For these reasons, it seems this paper is worthy of publication.

With that said, I have one concern that I hope the authors will address in their manuscript.

The concern is over the authors' interpretation of their laboratory experiments. In particular, how does the decision making process of their experimental subjects relate to how people actually behave in social settings, such as those involving cooperation.

My concern is that it is not obvious to me that the processes that shape learning in real world social settings resembles the process of learning in these experimental games.

That is, in the real world, people are sometimes consciously deliberating over strategies, but they are just as often acting on gut reactions; emotions, or ideologies. Human learning sometimes involves calculating best responses, but also involves social imitation, reinforcement learning, and evolution by natural selection. Of course this learning process can be simplified and modeled, but it isn't obvious to me why human subjects looking at abstract payoffs on a computer screen are informative about learning behavior in the real world.

In particular, the fact that the subjects in this experiment were likely to get stuck on local optima when the best responses are made salient, may be because the specific decision making process tapped into in this design is very susceptible to demand effects and saliency effects. Perhaps in the real world, evolutionary processes are less susceptible to these concerns, because evolution is influenced by lifetime payoffs? Perhaps social learning processes get around these concerns by allowing those who have reached global optima to cluster, and then be imitating by other clusters?

I hope the authors can add some discussion that directly addresses this point or discuss the limited inferences we can draw from such experiments.

Reviewers' comments and authors' responses

Reviewer #1 (Remarks to the Author) **and authors' responses in bold letters**

The authors conduct a two-player repeated game experiment to examine the consequences that two different payoff information scenarios for the participants have on their individual behaviour. In the first scenario (called NoInfo), participants are told the choices (in the interval [0.1, 6]) and payoffs of the two players in the previous period, while in Info they are also shown graphically what their payoff would have been in the previous round for all their choices given the actual choice of their partner.

The results show that participants tend to follow a myopic best reply heuristic (BR) leading to the unique Nash equilibrium (NE) of the one-shot game when the necessary information is available (i.e. in Info). However, in NoInfo, participants tend to follow an Imitate-the-Best heuristic for the first 25 periods or so and, over the remainder of the 600 periods, switch to a combination of adopting their partner's previous choice (Match) and adjusting their choice in the same (respectively, opposite) direction if their payoff increased (respectively, decreased) in the previous period (WCLR).

The game's payoff function is such that Imitate-the-Best leads to choice 6 with lowest payoff for both players, the unique NE is choice 3 with intermediate payoff, and Match combined with WCLR leads to choice 0.1 with highest payoff. Thus, the extra payoff information in Info produces a less cooperative outcome in the long-run, hence the title for the paper. For a different payoff function, one can well imagine the above behavioural heuristics would produce the opposite title.

The properties of the one-shot payoff function, now briefly mentioned on page 7, need to be discussed earlier in the paper. This will allow readers to understand such statements on page 4 as Imitate-the-Best copies "a higher action and made higher profits" and "pushes behaviour towards extremely competitive outcomes".

That is a good point. The payoff function has been moved to the first paragraph of the results section on page 2.

Moreover, more discussion of reference 24 (Friedman et al., 2015) and its connection to the current paper must be provided in the main text since its two-player experiment seems to be based on the Info treatment with twice as many (i.e. 1200) periods.

We added a crisp section how our paper is tied to reference 24 in the first paragraph of the results section on page 3.

The data the authors have on 1200 and 2400 period games should be given in an Appendix.

The data for both 1200 and 2400 period treatments have been included in the Supplementary Information, see Supplementary Figures 1 and 2 and the Supplementary Discussion.

More statistics should be provided for Figure 2 (e.g. means and variances)

We now included means and standard deviations for each phase in the figure legends, including the 1200 and 2400 period treatments in the Supplementary Information.

as well as extra columns in Table 1 for Periods 26-300 and Periods 300-600. Again, these can be relegated to an Appendix if necessary.

Initially we did not include extra columns as the separate results the learning phase (periods 26-300) and the long run phase (periods 300-600) are qualitatively similar to the pooled results and we figured it would be easier for the reader to digest two instead of three columns. Obviously, this is not fully consistent with the rest of the analysis where we analyze learning and long run separately. Therefore, we included Supplementary Table 1 in the Appendix showing the results for the two phases separately as by the referee's wishes.

Also, in Table 1, it is unclear to me whether the first four rows refer to data from only the treatment NoInfo. If the data is for the combined NoInfo and Info treatments, I think extra rows are needed for only data from NoInfo.

It seems that we did not state our variable definitions clearly enough. In our regression we pool data for NoInfo and Info treatments. The first four regressors (COPY UP, COPY DOWN, WCLR, BR) show results for the baseline (NoInfo). INFO is a dummy for the Info treatment, and is then interacted with all four regressors to pick up the difference between NoInfo and Info. These interaction terms we denote by COPY UP:INFO, COPY DOW:INFO and so on. Thanks for pointing this out. We included a note in the table's legend.

Overall, the paper shows clearly that details of payoff information available to the players has important consequences in predicting their behaviour, a valuable lesson given the recent proliferation of experiments based on more complicated games. I would like to see a revision that addresses my concerns above, including the relationship with reference 24.

Minor Comments. P4: Could mention "Myopic Best Reply" is only available in Info.

We included a line on that at the end of the first paragraph of the Individual Behavior section on page 6.

P4, line 11-: How does "Match align subjects' actions" rather than produce a two-cycle?

It is true that the matching heuristic when applied mechanically would produce a cycle. In our regressions we do not model Match like that though, but model it as a probability of changing one's action in the direction of the other subject's action. If steps are not too big this aligns subjects' actions. Therefore, we left the section unchanged.

P6, line 1-: Perhaps "unchanging over time" is better than "stable over time"?

We changed it to "does not change over time".

Reviewer #2 (Remarks to the Author) **and authors' responses in bold letters**

Huck et al. has subjects play a many round repeated game with a continuous action space where subjects learn their payoffs from each round as well as the payoffs the other player receives, and depending on the treatment, also learn the payoffs they could have received for every action they could have taken. Subject of course do not know the game form, i.e. the complete mapping between the action profile and the payoffs. The authors find the learning models that best fit the subjects data, and also analyze the long run behavior of subjects in each treatment, and convincingly demonstrate that subjects in the treatment with less payoff information perform better, presumably because they are less likely to get stuck in the local optima that consists of the Nash equilibrium of a single stage in the game but more likely to achieve the global optima represented by a more cooperative equilibria of the repeated game, presumably because when the subjects have more information they are more likely to employ as their learning process the heuristic of best response dynamic.

I find this result to be compelling, and the design to be rather clean, and fits in well with the learning and games literature, and also seems to fit a generally valid intuition that having immediate payoffs can make it more likely to get stuck at a local optima and prevent people from realizing long term gains, which can be crucial in repeated games settings. For these reasons, it seems this paper is worthy of publication.

With that said, I have one concern that I hope the authors will address in their manuscript.

The concern is over the authors' interpretation of their laboratory experiments. In particular, how does the decision making process of their experimental subjects relate to how people actually behave in social settings, such as those involving cooperation.

My concern is that it is not obvious to me that the processes that shape learning in real world social settings resembles the process of learning in these experimental games.

That is, in the real world, people are sometimes consciously deliberating over strategies, but they are just as often acting on gut reactions; emotions, or ideologies. Human learning sometimes involves calculating best responses, but also involves social imitation, reinforcement learning, and evolution by natural selection. Of course this learning process can be simplified and modeled, but it isn't obvious to me why human subjects looking at abstract payoffs on a computer screen are informative about learning behavior in the real world.

In particular, the fact that the subjects in this experiment were likely to get stuck on local optima when the best responses are made salient, may be because the specific decision making process tapped into in this design is very susceptible to demand effects and saliency effects. Perhaps in the real world, evolutionary processes are less susceptible to these concerns, because evolution is influenced by lifetime payoffs? Perhaps social learning processes get around these concerns by allowing those who have reached global optima to cluster, and then be imitating by other clusters?

I hope the authors can add some discussion that directly addresses this point or discuss the limited inferences we can draw from such experiments.

External validity is indeed an important point. We reflected upon these questions and added some discussion on the issue in our discussion on pages 8 and 9.

Reviewer #1 (Remarks to the Author):

The authors' revisions have adequately addressed my concerns with the original submission. My only new comment is connected with the new Supplementary Figures 1 (2 second periods) and 2 (4 second periods) and their comparison with Figure 2 (8 second periods) of the main text. Although the main points are still clear in all these figures that there are three phases to the experimental behavior patterns (initial phase, learning phase, and long-run phase) where Info differs from NoInfo and that less information leads to more cooperative behavior in the long-run, Info in the two Supplementary Figures does not seem to evolve to the Nash outcome as it does in the main text figure. A brief discussion of this is in order in the Appendix, especially if the authors can provide some insight into this difference. I also recommend minor changes to the main text, either to expand on the "similar results" on page 2 (line 43) or to refer to the Supplementary Figures on page 5 (line 93) where it is stated that "average action is indistinguishable from the Nash outcome".

The authors might also consider rewording the sentence on page 9 (lines 177 to 181). Currently, it is hard to read and difficult to understand.

Subject to these minor revisions, I recommend the paper be accepted for publication. The authors have clearly shown that details of payoff information available to players have important consequences in predicting their behavior, a valuable lesson given the recent proliferation of game experiments examining the evolution of cooperation.

Reviewer #2 (Remarks to the Author):

I recommend publication of the revised version.

Reviewers' comments and **authors' responses in bold letters**

Reviewer #1 (Remarks to the Author):

The authors' revisions have adequately addressed my concerns with the original submission. My only new comment is connected with the new Supplementary Figures 1 (2 second periods) and 2 (4 second periods) and their comparison with Figure 2 (8 second periods) of the main text. Although the main points are still clear in all these figures that there are three phases to the experimental behavior patterns (initial phase, learning phase, and long-run phase) where Info differs from NoInfo and that less information leads to more cooperative behavior in the long-run, Info in the two Supplementary Figures does not seem to evolve to the Nash outcome as it does in the main text figure. A brief discussion of this is in order in the Appendix, especially if the authors can provide some insight into this difference.

The referee is correct that, in the long-run, behaviour in the Info treatments in the 2 seconds and 4 seconds treatments does not stick perfectly to Nash, even though the treatment difference still is significant. We added some discussion in the Appendix.

I also recommend minor changes to the main text, either to expand on the "similar results" on page 2 (line 43) or to refer to the Supplementary Figures on page 5 (line 93) where it is stated that "average action is indistinguishable from the Nash outcome".

That is a good point. We added a reference to the Supplementary Figures.

The authors might also consider rewording the sentence on page 9 (lines 177 to 181). Currently, it is hard to read and difficult to understand.

After rereading that sentence we agree with the referee. We slightly changed the structure to make it easier to understand.

Subject to these minor revisions, I recommend the paper be accepted for publication. The authors have clearly shown that details of payoff information available to players have important consequences in predicting their behavior, a valuable lesson given the recent proliferation of game experiments examining the evolution of cooperation.

Thank you for the many helpful comments.

Reviewer #2 (Remarks to the Author):

I recommend publication of the revised version.

Thank you also for the helpful comments in the first round.